# SToRe3D: Sparse Token Relevance in ViTs for Efficient Multi-View 3D Object Detection

## Abstract

Vision Transformers (ViTs) enable strong multi-view 3D detection but are limited by high inference latency from dense token and query processing across multiple views and large 3D regions. Prior sparsity methods, designed mainly for 2D vision, prune or merge image tokens but do not extend to full-model sparsity or address 3D object queries. We introduce SToRe3D, a relevance-aligned sparsity framework that jointly selects 2D image tokens and 3D object queries while storing filtered features for selective reuse. Mutual 2D–3D relevance heads allocate compute to driving-critical content and preserve other embeddings. Evaluated on nuScenes and our new nuScenes-Relevance benchmark, SToRe3D delivers up to 3× faster inference with marginal accuracy loss, establishing real-time 3D detection with large scale ViTs while maintaining accuracy on planning-critical agents.

## 1 Introduction

Transformers dominate modern perception, yet dense attention over long image sequences and large 3D search spaces remains a barrier to real-time deployment. In autonomous driving, where latency and safety are essential, the challenge is not only to *reduce* computation but to *allocate* it selectively, raising the question: how can we focus compute only on content that matters for decision-making?

Vision Transformer (ViT) (Dosovitskiy et al., 2020) backbones and Detection Transformer (DETR) (Carion et al., 2020) decoders achieve strong 3D perception but incur quadratic costs over tokens and queries. Yet urban scenes are dominated by background (sky, road, buildings) and agents inconsequential for motion planning. Uniform computation is thus wasteful, treating all tokens and candidate objects as equally important and misaligning perception with the downstream planner.

Prior efficiency work focuses on one modality in isolation. ViT sparsity methods prune or merge *image tokens* for 2D tasks (Rao et al., 2021; Bolya & Hoffman, 2023; Liu et al., 2024; Huang et al., 2025), while DETR variants suppress subsequent *encoder tokens* or *decoder queries* for 2D detection (Roh et al., 2021; Zheng et al., 2023). For multi-view 3D detection, ToC3D (Zhang et al., 2024) compresses tokens using historical queries, reducing backbone cost but leaving query redundancy and planner relevance under-exploited. None of these approaches enforce end-to-end sparsity over both 2D tokens and 3D queries, or align it with planner relevance.

We introduce SToRe3D, a *planner-aligned* sparsity framework that scores and routes both image tokens and 3D object queries using lightweight mutual 2D–3D relevance heads. High-relevance items are processed by deeper layers, while lower-relevance embeddings are not discarded but *stored* in feature buffers for selective reactivation. This *store*–reactivation design avoids the merge–unmerge overhead of token compression approaches, applies on the *first frame* (no history required), and yields end-to-end gains by reducing both $\mathcal{O}(N^2)$ self-attention in the backbone and decoder.

Our contributions are: **(i) Unified end-to-end sparsity** that *jointly* prunes tokens and queries across a single architecture. **(ii) Planning-aligned relevance** supervised by *future interaction corridors* capturing short-horizon ego–agent proximity, aligning perception budgets with planning. **(iii) Real-time ViTs at scale** via store–reactivate buffers that preserve recoverability for aggressive sparsity, enabling further latency reduction. **(iv) nuScenes-Relevance** (nuScenes-R), a benchmark that measures accuracy specifically on planning-critical agents. SToRe3D reduces latency by up to 3× with marginal accuracy loss, delivering real-time throughput (up to 18 FPS) for medium and large ViTs and providing a practical path to planner-aware efficiency in multi-view 3D detection.

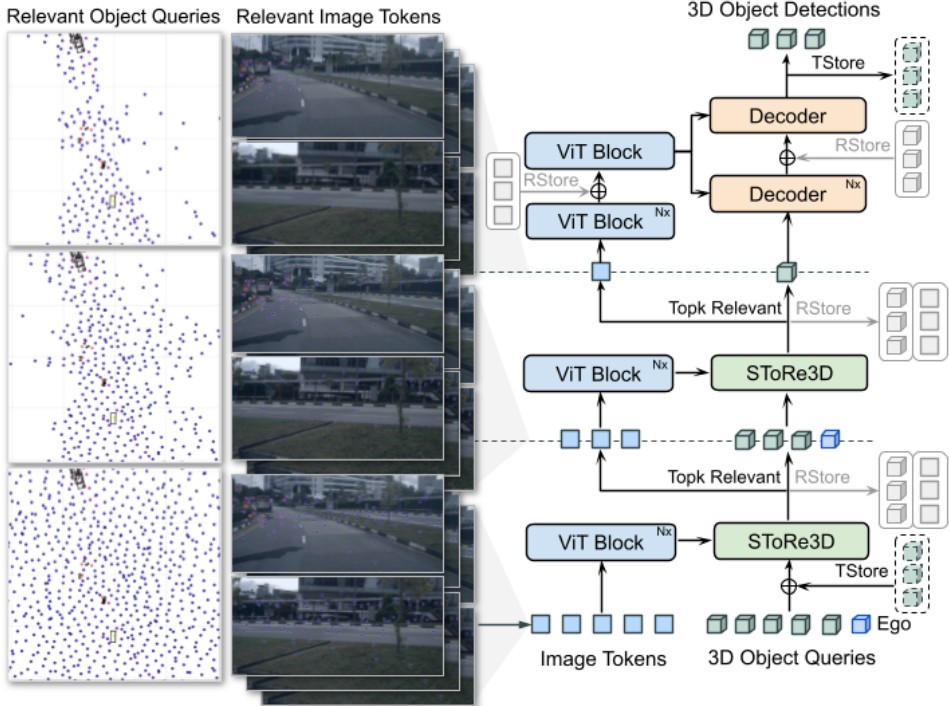

Figure 1: SToRe3D routes computation via planning-aligned relevance. Tokens and queries above stage-wise thresholds are processed deeper while the rest are *stored* for reactivation.

## 2 RELATED WORK

**Multi-view 3D object detection.** Early camera-only 3D detectors lifted multi-view features into BEV space before aggregation (Philion & Fidler, 2020). Transformer-based methods such as DETR3D (Wang et al., 2022) and PETR (Liu et al., 2022b) introduced 3D queries to attend across views, while BEVFormer (Li et al., 2022c), Sparse4D (Lin et al., 2022), and StreamPETR (Wang et al., 2023b) improved accuracy and efficiency with deformable attention and temporal aggregation. Yet real-time deployment with large ViTs remains challenging, as covering wide 3D regions still requires processing a vast number of tokens and queries. We address this bottleneck through planner-aligned sparsity applied jointly to both modalities.

**ViT token sparsity.** Reducing transformer attention cost has been explored through approximate attention (Choromanski et al., 2020; Wang et al., 2020; Dao et al., 2022; Ainslie et al., 2023), component pruning (Voita et al., 2019; Michel et al., 2019; Meng et al., 2022), and vision-specific inductive biases (Mehta & Rastegari, 2021; Graham et al., 2021; Liu et al., 2021). For ViTs, token *pruning* approaches learn saliency to drop patches progressively (Rao et al., 2021; Liang et al., 2022; Xu et al., 2022; Fayyaz et al., 2022; Yao et al., 2022; Li et al., 2022b; Yu et al., 2024; Xu et al., 2023), *learned tokenization* approaches select informative latent tokens (Ryoo et al., 2021; Tang et al., 2022), and *merging/fusion* approaches reduce redundancy by associating similar tokens (Bolya & Hoffman, 2023; Xu et al., 2024; Lee & Hong, 2024; Lee et al., 2024). Extensions to dense tasks such as detection exist (Liu et al., 2024; Huang et al., 2025). However, these methods operate on *image tokens only*, assume 2D salience, and provide no mechanism to coordinate with 3D object queries, precisely required for multi-view 3D detection.

**DETR token sparsity.** Efficiency in DETR-style detectors is typically achieved by sparsifying encoder tokens or decoder queries. Deformable DETR replaces global attention with sparse, reference-point sampling (Zhu et al., 2020), while Sparse-DETR, Focus-DETR, and Salience-DETR further limit token or region updates through learned salience (Roh et al., 2021; Zheng et al., 2023; Hou et al., 2024). Orthogonal variants such as DN-/DAB-/DINO-/RT-DETR (Li et al., 2022a; Liu et al.,

2022a; Zhang et al., 2022; Zhao et al., 2024; Yao et al., 2021) accelerate convergence mainly via query initialization and denoising rather than structural sparsity. For multi-view 3D detection, Focal-PETR selects foreground tokens for the decoder with a 2D auxiliary head (Wang et al., 2023a), while ToC3D compresses backbone tokens using history-driven scores and merge–unmerge routing (Zhang et al., 2024). Yet these approaches remain largely *token-only*, focusing on the backbone or encoder while leaving decoder query redundancy under-exploited. ToC3D's dependence on temporal priors also limits first-frame efficiency and its per-block regrouping introduces extra overhead. In contrast, SToRe3D applies *joint* 2D–3D sparsity, works from the first frame, and avoids merge–unmerge complexity via lightweight *store–reactivate* buffers.

**Planning- and safety-critical perception.** Beyond efficiency, perception works explicitly focuses on agents that matter for decision-making. Examples include risk-object identification (Li et al., 2020; 2023a), spatial attention guided by planning (Wei et al., 2021), and end-to-end planning–perception frameworks such as UniAD (Hu et al., 2023) and SparseDrive Sun et al. (2025). While these methods couple perception to downstream planning goals, they rely on CNN backbones and lack a general *architectural* mechanism for scalable token and query sparsity to enable ViTs. SToRe3D closes this gap by supervising sparsity with a *future interaction corridor* and benchmarking with nuScenes-R, achieving end-to-end latency gains while focusing on planning-critical agents.

## 3 METHOD

SToRe3D applies *joint, hierarchical sparsity* to both image tokens and 3D object queries in a temporal multi-view 3D detector Wang et al. (2023b). At each stage, lightweight relevance heads score tokens and queries. High scores are processed deeper, while low scores are written to *storage buffers* for selective reuse. This store–reactivate design reduces $\mathcal{O}(N^2)$ self-attention while avoiding irreversible pruning.

### 3.1 PROBLEM FORMULATION

We consider multi-view 3D detection with $V$ synchronized cameras over a temporal window $\{t-T, \ldots, t\}$. Each view produces tokens $\mathbf{X}_{t,v}$ from a ViT backbone, concatenated as $\mathbf{X}_t$. The backbone interleaves global and windowed attention and a feature pyramid network (FPN) provides multi-scale features. Detection object queries $\mathbf{Q}_t$ are anchored at 3D positions $\mathbf{p} = (x, y, z)$, initialized as $\mathbf{q}^{(0)} = \mathrm{MLP}(\mathrm{PE}(\mathbf{p}))$, and refined via a deformable DETR-style decoder.

Following streaming detectors (Wang et al., 2023b; Lin et al., 2023), top-$K$ queries propagate across frames, maintained in a temporal memory with temporal reference points that transformed to current ego frame. The active query set combines propagated and initialized queries, $N_d = N_{\mathrm{prop}} + N_{\mathrm{init}}$; at the first frame additional initialized queries replaces the propagated queries. This yields unified token $\mathbf{X}_t$ and query $\mathbf{Q}_t$ sets, input to the sparse relevance module.

### 3.2 DEFINING OBJECT RELEVANCE

We align sparsity with planning using a *future interaction corridor* in BEV. Let $\mathcal{B}_{\mathrm{ego}}(\tau)$ and $\mathcal{B}_i(\tau)$ denote oriented boxes for ego and agent-$i$ at $t + \tau$. Swept sets are defined by the convex hull of the unions of future boxes:

$$\mathcal{S}_{\mathrm{ego}}(H) = \mathrm{conv}\Big( \bigcup_{\tau \in [0,H]} \mathcal{B}_{\mathrm{ego}}(\tau) \Big), \quad \mathcal{S}_i(H) = \mathrm{conv}\Big( \bigcup_{\tau \in [0,H]} \mathcal{B}_i(\tau) \Big). \tag{1}$$

An agent is *relevant* if the closest distance between its swept polygon and the ego swept polygon is within a safety margin, $d_{\mathrm{min}}$:

$$y_i^{\mathrm{rel}} = \mathbb{1}\Big( \mathrm{dist}\big( \mathcal{S}_i(H), \mathcal{S}_{\mathrm{ego}}(H) \big) \leq d_{\mathrm{min}} \Big). \tag{2}$$

This polygonal corridor captures translation and orientation over discrete *future* steps for $H{=}5$ seconds where the labels $\{y_i^{\mathrm{rel}}\}$ supervise relevance. These same definitions apply to nuScenes-R metrics (Sec. 4).

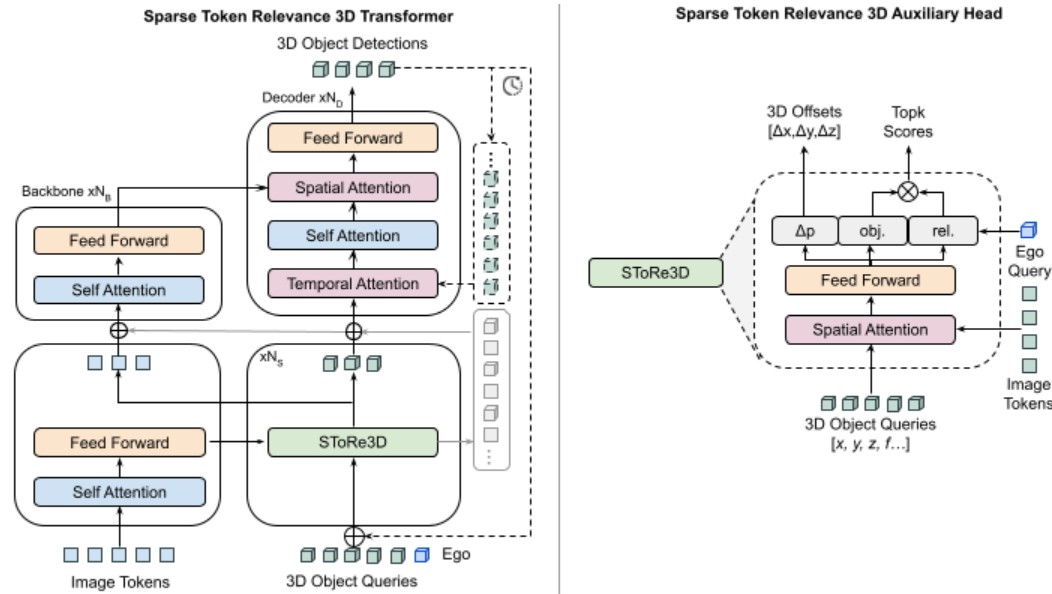

Figure 2: SToRe3D architecture with ViT backbone and transformer decoder. Relevance heads apply stage-wise store–reactivate on tokens and queries.

## 3.3 Unified 2D–3D relevance prediction

We predict planning-aligned relevance for both modalities using *mutual gating*: queries are scored in the context of tokens and vice versa. Query relevance is supervised by corridor labels, while token relevance is aggregated from query attention. For object query $\mathbf{q}_j$, we compute a context vector from cross-attended tokens and optionally an ego embedding $\mathbf{e}_t$:

$$\mathbf{c}_j^{\text{qry}} = \text{CrossAttn}(\mathbf{q}_j, \mathbf{X}_t) \oplus \mathbf{e}_t, \quad r_j^{\text{qry}} = \sigma\Big(\mathbf{u}^\top \phi([\mathbf{q}_j \,\|\, \mathbf{c}_j^{\text{qry}}])\Big). \tag{3}$$

where $\phi$ is a small MLP, $\oplus$ indicates optional concatenation, and $\sigma$ is the sigmoid function. The ego term lets $r^{\text{qry}}$ condition relevance on the ego–agent motion. Image token relevance $r_i^{\text{img}}$ aggregates attention from top-$K$ relevant queries:

$$r_i^{\text{img}} = \frac{1}{K} \sum_{j \in \mathcal{K}^{\text{qry}}} A_{j \to i}, \tag{4}$$

yielding a query-aware token relevance that emphasizes regions supported by high-relevance 3D queries. The scores $r_j^{\text{qry}}$ and $r_i^{\text{img}}$ serve as routing signals for stage-wise sparsification (Section 3.4). We supervise $r^{\text{qry}}$ with binary labels $y^{\text{rel}}$ from the interaction corridors (Section 3.2).

## 3.4 Hierarchical Token Storage

Joint sparsity is applied to both the *backbone token stream* and the *query stream* within the backbone and encoder. After each stage $\ell$, we *filter* tokens/queries using the relevance scores from Section 3.3, *store* the remainder in buffers, and optionally *reintroduce* a small subset at deeper stages. For stage-wise filtering and storage let $N_\ell$ and $Q_\ell$ be the numbers of tokens and queries at stage $\ell$. We keep fractions $\rho_\ell^{\text{img}}, \rho_\ell^{\text{qry}} \in (0, 1]$ via Gumbel-softmax Topk (Jang et al., 2016) and the filtered items are written to buffers.

$$\mathcal{K}_\ell^{\text{img}} = \text{TopK}(\mathbf{r}_\ell^{\text{img}}, \lfloor \rho_\ell^{\text{img}} N_\ell \rfloor), \quad \mathcal{K}_\ell^{\text{qry}} = \text{TopK}(\mathbf{r}_\ell^{\text{qry}}, \lfloor \rho_\ell^{\text{qry}} Q_\ell \rfloor). \tag{5}$$

The fractions $\rho_\ell^{\text{img}}, \rho_\ell^{\text{qry}} \in (0, 1]$ follow a non-increasing *hierarchical schedule* with depth and are regularized toward targets in Section 3.5. Let $\overline{\mathcal{K}}_\ell^{\text{img}}$ and $\overline{\mathcal{K}}_\ell^{\text{qry}}$ be complements of the kept indices. We write filtered features to buffers immediately after filtering and before the next stage:

$$\mathbf{S}_\ell^{\text{img}} \leftarrow \mathbf{X}_\ell[\overline{\mathcal{K}}_\ell^{\text{img}}], \qquad \mathbf{S}_\ell^{\text{qry}} \leftarrow \mathbf{Q}_\ell[\overline{\mathcal{K}}_\ell^{\text{qry}}]. \tag{6}$$

To mitigate early information loss, we allow *reactivation*: at stage $\ell+1$ we recompute scores $\widehat{\mathbf{r}}$ on buffered items using the updated context and reinsert up to $\kappa_\ell^{\mathrm{img}}$ tokens and $\kappa_\ell^{\mathrm{qry}}$ queries. Gradients flow through any reintroduced items. We use a two-level storage schedule: (i) *depth-wise* budgets $(\rho_\ell^{\mathrm{img}}, \rho_\ell^{\mathrm{qry}})$ are non-increasing with $\ell$; (ii) *training-time* pruning is introduced gradually (linear warm-up from dense to target budgets) to avoid optimization shocks. For additional robustness under aggressive sparsity, we also enable a *last-layer reactivation* pass to recover stored items. This preserves global context without restoring all pruned items and adds negligible cost.

### 3.5 OPTIMIZATION APPROACH

The overall framework is trained using a multi-task objective to optimize both detection and relevance learning simultaneously. For detection, a combination of focal loss Lin (2017) for classification and L1 loss for bounding box regression is used with Hungarian bipartite matching. For relevancy, a binary cross entropy classification loss is used between predicted scores ($r^{\mathrm{qry}}$) and corridor-derived labels $y^{\mathrm{rel}}$. In addition, an auxiliary loss is used to supervise ROI feature extraction which also contains a classification and regression loss on targets in the 2D image space Wang et al. (2023a). The joint loss function is formulated as:

$$\mathcal{L} \;=\; \mathcal{L}_{\mathrm{det}} + \lambda_{\mathrm{rel}}\mathcal{L}_{\mathrm{rel}}^{\mathrm{qry}} + \lambda_{\mathrm{aux}}\mathcal{L}_{\mathrm{aux}} \tag{7}$$

where $\lambda_{\mathrm{rel}}$ and $\lambda_{\mathrm{aux}}$ are balancing weights. $\mathcal{L}_{\mathrm{det}}$ includes focal and L1 losses with Hungarian matching. Query relevance uses Gaussian focal loss Law & Deng (2018) with corridor labels. Token relevance uses is indirectly supervised from cross-attention with the queries. Gumbel-TopK provides differentiable routing, with pruning linearly increased over training iterations from dense (no sparsity) to target sparsity budgets.

## 4 BENCHMARKING RELEVANCE (NUSCENES–R)

**Why relevance?** Conventional detectors expend equal compute on all agents, inflating latency and misaligning perception with planning. Many urban objects (e.g., parked vehicles, distant pedestrians) are inconsequential for planning. We therefore evaluate perception under a *relevance-driven* lens: prioritize agents that matter for near-term driving. To quantify sparsity headroom, we vary the number of detected agents provided to a learned planner. Performance saturates with only 10–20 agents (Figure 3a), indicating substantial room for compute reduction without harming planning.

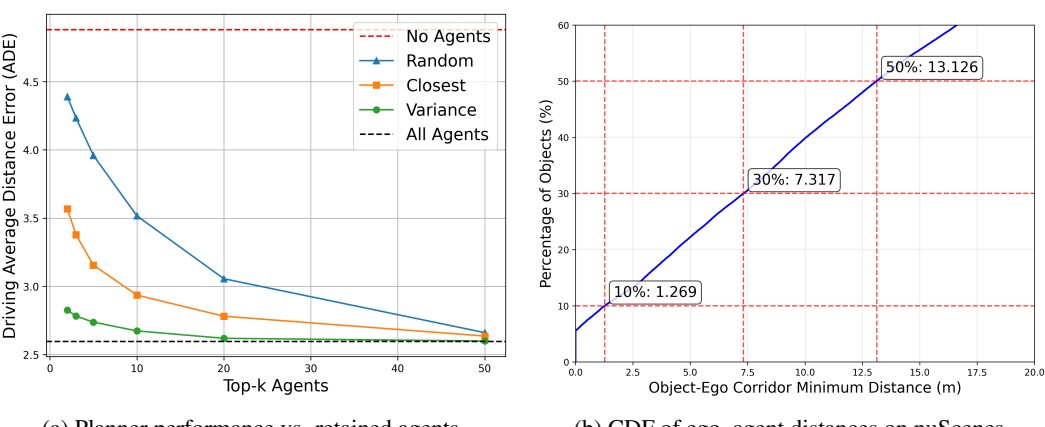

(a) Planner performance vs. retained agents.  (b) CDF of ego–agent distances on nuScenes.

Figure 3: Focusing on a small subset of agents suffices for planning, motivating nuScenes–R.

**Planning-relevant object labels.** Ground-truth relevance follows the future interaction corridors of ego and other agents (Sec. 3.2). Swept sets $\mathcal{S}_{\mathrm{ego}}(H), \mathcal{S}_i(H)$ are buffered by $d_{\mathrm{min}}$, and agents within this margin are labeled $y_i^{\mathrm{rel}} = 1$. To fix a single operating point across scenes, $d_{\mathrm{min}}$ is chosen by percentile of the ego–agent distance distribution (Fig. 3b); in results we use the 10th

percentile, yielding $d_{min}$=1.269 meters over $H$=5 seconds and $\sim$3 relevant agents per frame (max $\sim$30). Labels are generated offline by discretizing trajectories, convexifying swept polygons, and applying the buffered-intersection test.

**Relevance metrics.** Standard detection metrics mean average precision(mAP) and nuScenes detection score (NDS) treat all agents equally, regardless of planning importance. We instead define relevance via a *future interaction corridor*: 5-second swept polygons for ego and each agent. An agent is labeled relevant if the closest distance $d_C$ between its corridor and ego's corridor is below a buffer $d_{RM}$. Empirically, $d_{RM}$=1.2 m (the 10th percentile of ego–agent distances) selects $\sim$10% of agents—about three per frame on average, with a maximum of 31. This definition underlies our benchmark, nuScenes-R.

We report two variants: *relevant motion* (RM) filtered metrics (mAP-RM, NDS-RM), which apply the RM filter as described above, and *relevant area* filtered metrics (mAP-RA, NDS-RA), which use a fixed detection area around the vehicle for evaluation. Because RM relies on privileged future information, evaluation is done in two-passes: (i) detections are matched to RM-filtered ground truth for true positives and false positives; (ii) false negatives are counted against the full set. This preserves nuScenes protocol while avoiding unfair penalties on deprioritized agents. Together, RM and RA ensure that SToRe3D's relevance-adaptive sparsity is evaluated fairly, measuring whether accuracy is preserved on *planning-critical* agents while enabling substantial efficiency gains.

# 5 EXPERIMENTS

## 5.1 EXPERIMENT SETUP

**Dataset.** We evaluate on the nuScenes 3D detection benchmark (Caesar et al., 2020), which contains 1,000 $\sim$ 20s scenes at 20 Hz with six surround cameras per sample. Camera intrinsics and extrinsics are provided. Annotations are available every 0.5 s, yielding 28k/6k/6k annotated samples for train/val/test across ten classes (vehicles, pedestrians, cyclists, etc.). We report standard nuScenes metrics mAP and NDS. Because these metrics weight all agents equally, we additionally evaluate with relevance-filtered metrics on planning-critical agents using our nuScenes-R protocol (Section 4; Figure 3), including Relevant-Area (RA) and Relevant-Motion (RM) variants aligned with the future interaction corridor.

**Implementation Details.** We evaluate on nuScenes using six synchronized cameras with standard intrinsics/extrinsics. Backbones include ResNet-50/101 (He et al., 2016), V2-99 (Lee et al., 2019), and ViT-B/L (Dosovitskiy et al., 2020), initialized from NuImages (Caesar et al., 2020), DD3D (Park et al., 2021), and EVA-02 (Fang et al., 2024) respectively. Input resolution is $320 \times 800$ unless otherwise noted; we also report $256 \times 704$, $512 \times 1408$, and $800 \times 1600$ for accuracy–speed trade-offs. The detector follows a DETR-style design with multi-scale features, $D$=256 embedding dimension, and $L$=6 decoder layers. As a dense baseline we use 644 detection queries and 256 temporal queries (900 total), with four frames of memory (1024 queries). Denoising (Wang et al., 2023a;b) is applied during training. Models are trained for 24 epochs on 8$\times$A100 GPUs (batch size 16). Inference latency is measured at batch size 1 on a single RTX3090. Optimization uses AdamW with cosine decay, gradient clipping, and mixed precision. Relevance heads are two-layer MLPs with GELU; TopK gating employs a differentiable Gumbel-softmax. We report three operating points, SToRe3D-1/2, SToRe3D-1/4, and SToRe3D-1/10, corresponding to hierarchical schedules that retain roughly half, quarter, and tenth of tokens/queries. Further hyperparameters appear in the appendix.

## 5.2 MAIN RESULTS

**Accuracy–efficiency trade-offs.** Figure 4 plots the speed–accuracy frontier of SToRe3D across sparsity regimes, alongside StreamPETR (Wang et al., 2023b). Jointly pruning 2D tokens and 3D queries yields monotonic FPS gains with negligible accuracy loss at low sparsity and only small losses at higher sparsity. Notably, SToRe3D-1/10 establishes *real-time* ViT-based multi-view 3D detection, running at $\sim$18 FPS with ViT-B while remaining SOTA among methods at similar latency. To the best of our knowledge the first time transformer models of this scale reach real-time

on nuScenes. As the hierarchical keep-ratio decreases, inference time drops monotonically while mAP/NDS degrade smoothly, with a clear knee at mid-sparsity.

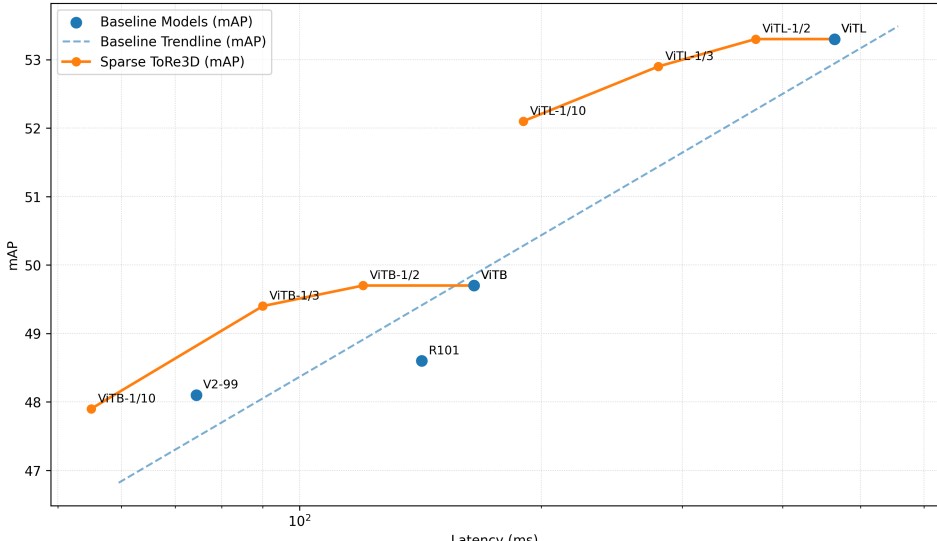

Figure 4: Latency-accuracy curves for SToRe3D under varying sparsity.

**Comparison to baselines.** Table 1 compares SToRe3D with strong vision baselines at comparable latency levels on nuScenes and nuScenes-R. SToRe3Dachieves competitive accuracy while delivering materially higher FPS than dense or token-only sparse baselines. Relative to StreamPETR and ToC3D, joint token–query sparsity improves the accuracy–latency balance, confirming that joint pruning is more effective than token-only compression.

On nuScenes-R, where metrics are restricted to agents within the future interaction corridor based on Relevant Motion (RM) and Relevance Area (RA) filtered metrics. SToRe3D retains strong mAP-RM/NDS-RM on planning-critical agents while delivering higher FPS than dense baselines (Stream-PETR), indicating that compute is effectively redirected to the most decision-relevant content. SToRe3D-1/10, with its design explicitly tailored for downstream driving applications, achieves 0.521 mAP, 0.607 NDS, 0.278 mAP-RM, 0.478 NDS-RM, and 5.2 FPS. Compared to the baseline model StreamPETR, which records 0.523 mAP-RA, 0.610 NDS-RA, 0.264 mAP-RM, 0.463 NDS-RM, and 2.7 FPS, SToRe3D demonstrate superior performance in prioritizing and accurately detecting relevant objects within the future interaction corridor. This underscores the effectiveness of SToRe3D's relevance-driven sparsity in delivering high performance where it matters most for safety-critical autonomous systems.

### 5.3 ABLATION STUDY

**Sparsity approach.** Table 2 compares alternative sparsification strategies for scoring of image tokens at matched keep ratios against our joint token-query sparsity. At $\rho=0.5$ (top block) and $\rho=0.3$ (bottom block), SToRe3D consistently attains equal or higher accuracy at similar or lower latency than token-only approaches. The additional speedup arises primarily from query sparsity in the decoder, which token-only methods cannot realize, while tying token pruning to object queries preserves accuracy under stronger sparsity.

**Pruning design.** Table 3 provides am ablation study of various design decisions within the SToRe3D framework, evaluating their impact on performance. We find that joint pruning of image tokens *and* object queries (I&O) outperforms pruning either stream alone in terms of latency reduction. Retaining filtered items in *store* buffers with optional reactivation is superior to hard *pruning*, indicating that retrieval paths mitigate early pruning errors. Finally, a linear schedule (warm up from dense to target $\rho$) improves stability over flat query reduced fine-tuning.

| Methods | Backbone×W-S | mAP↑ | -RA↑ | -RM↑ | NDS↑ | -RA↑ | -RM↑ | FPS↑ |
|---|---|---|---|---|---|---|---|---|
| PETRv2 | R50×704 | 0.349 | - | - | 0.456 | - | - | 20.8 |
| BEVDepth | R50×704 | 0.351 | - | - | 0.475 | - | - | 17.3 |
| StreamPETR | R50×704 | 0.449 | 0.569 | 0.227 | 0.546 | 0.617 | 0.529 | 35.2 |
| SToRe3D-1/10 | ViT-B×800-1/10 | 0.479 | 0.612 | 0.241 | 0.571 | 0.639 | 0.43 | 17.7 |
| BEVStereo | R50×704 | 0.372 | - | - | 0.5 | - | - | 13.4 |
| SOLOFusion | R50×704 | 0.427 | - | - | 0.534 | - | - | 12.5 |
| StreamPETR | V2-99×800 | 0.482 | 0.605 | 0.248 | 0.571 | 0.646 | 0.428 | 13.5 |
| SToRe3D-1/3 | ViT-B×800-1/3 | 0.489 | 0.623 | 0.246 | 0.578 | 0.65 | 0.435 | 10.6 |
| ToC3D-Faster | ViT-B×800-1/3 | 0.453 | 0.618 | 0.243 | 0.559 | 0.656 | 0.43 | 7.3 |
| StreamPETR | R101×1408 | 0.486 | - | - | 0.578 | - | - | 7.0 |
| SToRe3D-1/2 | ViT-B×800-1/2 | 0.493 | 0.627 | 0.247 | 0.581 | 0.665 | 0.441 | 8.2 |
| StreamPETR | ViT-B×800 | 0.497 | 0.627 | 0.247 | 0.584 | 0.667 | 0.443 | 6.1 |
| ToC3D-Fast | ViT-B×800-1/2 | 0.46 | 0.615 | 0.232 | 0.562 | 0.664 | 0.431 | 6.6 |
| SToRe3D-1/10 | ViT-L×800-1/10 | 0.521 | 0.641 | 0.278 | 0.607 | 0.679 | 0.478 | 5.2 |
| BEVFormer | R101-DCN×1600 | 0.416 | - | - | 0.517 | - | - | 3.3 |
| Sparse4D | R101-DCN×1600 | 0.436 | - | - | 0.541 | - | - | 4.7 |
| ToC3D-Faster | ViT-L×800-1/3 | 0.517 | 0.63 | 0.257 | 0.609 | 0.672 | 0.453 | 3.1 |
| SToRe3D-1/3 | ViT-L×800-1/3 | 0.523 | 0.654 | 0.275 | 0.609 | 0.678 | 0.469 | 3.5 |
| ToC3D-Fast | ViT-L×800-1/2 | 0.523 | 0.639 | 0.264 | 0.610 | 0.681 | 0.463 | 2.5 |
| StreamPETR | ViT-L×800 | 0.521 | 0.641 | 0.288 | 0.608 | 0.688 | 0.485 | 2.2 |
| SToRe3D-1/2 | ViT-L×800-1/2 | 0.533 | 0.666 | 0.286 | 0.618 | 0.697 | 0.475 | 2.7 |

Table 1: Detection performance on nuScenes and nuScenes-R validation set against SOTA methods with comparable latency. Backbone ×W and -S are the image width and sparsity level respectively. StreamPETR (Wang et al., 2023b), ToC3D (Zhang et al., 2024), PETRv2 (Liu et al., 2023), BEVDepth (Li et al., 2023c), BEVStereo (Li et al., 2023b), SOLOFusion (Park et al., 2022), BEV-Former (Li et al., 2022c), and Sparse4D (Lin et al., 2023) are as reported.

| Sparsity Approach | TKR | NDS ↑ | mAP ↑ | FPS ↑ |
|---|---|---|---|---|
| StreamPETR | 1 | 0.612 | 0.521 | 2.15 |
| + Random | 0.5 | 0.567 (-7.4%) | 0.465 (-10.7%) | 2.45 (1.14×) |
| + DynamicViT | 0.5 | 0.597 (-2.5%) | 0.505 (-3.1%) | 2.47 (1.15×) |
| + ToC3D3D-Fast | 0.5 | 0.61 (-0.3%) | 0.523 (0.4%) | 2.43 (1.13×) |
| + SToRe3D-1/2 | 0.5 | 0.618 (1.0%) | 0.533 (2.3%) | 2.70 (1.26×) |
| + Random | 0.3 | 0.485 (-20.8%) | 0.36 (-30.9%) | 2.9 (1.35×) |
| + DynamicViT | 0.3 | 0.593 (-3.1%) | 0.493 (-5.4%) | 2.92 (1.36×) |
| + ToC3D3D-Faster | 0.3 | 0.603 (-1.5%) | 0.512 (-1.7%) | 2.89 (1.34×) |
| + SToRe3D-1/3 | 0.3 | 0.609 (-0.5%) | 0.523 (0.4%) | 3.51 (1.63×) |
| + SToRe3D-1/10 | 0.1 | 0.607 (-0.8%) | 0.521 (0%) | 5.21 (2.42×) |

Table 2: Ablation of sparsity approaches Wang et al. (2023b); Rao et al. (2021); Zhang et al. (2024) at matched keep ratios.

## 5.4 QUALITATIVE RESULTS

Figure 5 qualitatively highlights SToRe3D's improved detection over baselines in autonomous driving scenarios. SToRe3D suppresses background false positives by pruning low-relevance tokens and maintains detections for agents inside the interaction corridor, reducing critical false negatives. This yields focused detections on planning-critical agents within the future interaction corridor leading to more accurate and reliable perception where it matters for driving.

| Setting | Train Loss | Pruner | Pruned | Schedule | mAP↑ |
|---|---|---|---|---|---|
| None | - | - | - | - | 0.540 |
| v1 | Top Q | Store | I & O | Finetune | 0.515 |
| v3 | Top Q | Store | O | Linear | 0.534 |
| v4 | Top Q | Store | I | Linear | 0.527 |
| v5 | Top Q | Remove | I & O | Linear | 0.495 |
| v6 | All Q | Store | I & O | Linear | 0.513 |
| SToRe3D-1/10 | Top Q | Store | I & O | Linear | 0.521 |

Table 3: Ablation of design decisions on nuScenes validation set using ViT-L backbone.

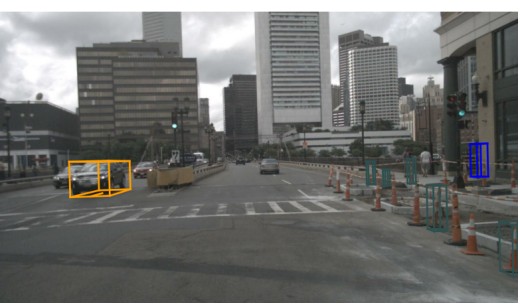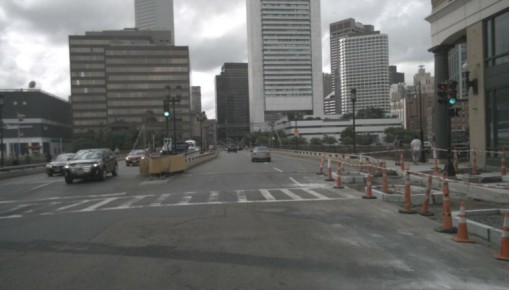

Figure 5: Vizualization of false negative cases for the baseline StreamPETR-R50 (left) and our SToRe3D-1/10-ViTB with similar latency (right).

## 6 CONCLUSION

We introduced SToRe3D, a planner-aligned sparsity framework for multi-view 3D detection with ViTs. SToRe3D applies joint, hierarchical pruning to both image tokens and 3D queries, replacing hard drops with filter and store buffers to allow selective reactivation. Relevance is supervised by a future interaction corridor and we proposed nuScenes-R to measure accuracy on planning-critical agents. On nuScenes, SToRe3D reduces latency by up to $3\times$ with marginal accuracy loss; at aggressive sparsity (SToRe3D-1/10) it reaches *real-time* throughput, $\sim 18$ FPS with ViT-L and near real-time $\sim 5$ FPS with ViT-B at $320\times800$, while achieving state of the art performance for methods with similar latency. Since nuScenes-R relies on corridor hyperparameters $(H, d_{\min})$ future work includes end-to-end relevance learning with planning, LiDAR sensor fusion, and closed-loop evaluation.

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

# Supplementary Material

## A   IMPLEMENTATION DETAILS

We follow standard multi-view 3D detection settings on nuScenes using 6 cameras, synchronized frames, and camera intrinsics/extrinsics. Backbones are ViT-based; we evaluate both medium and large variants. There is no encoder after the backbone and the decoder follows a DETR-style design with multi-scale features. We measure end-to-end latency at batch 1 with standard pytorch profilers on a single RTX3090 GPU. The ViT-L and ViT-B backbones follow EVA-02 (Fang et al., 2024) which was used for pretrained. The ViT-L backbone has 1024 embedding channels, 24 layers (6 global and 6 windowed attention), 16 attention heads, 0.3 drop path rate. The ViT-B backbone has 768 embedding channels, 12 layers (8 global and 16 windowed attention), 12 attention heads, 0.1 drop path rate. Both backbones use a patch size of 16 and window size of 16. The detection head uses 6 decoder layers with a 256 query embedding channels. The detailed token pruning ratios are provided in Table 4.

| Type | TKR | MKR | LKR0 | LKR1 | LKR2 | LKR3 | LKR4 | LKR5 | LKR6 | LKR7 |
|---|---|---|---|---|---|---|---|---|---|---|
| **ViT-L Models** | | | | | | | | | | |
| ToC3D-fast | 0.5 | 0.68 | 1.00 | 1.00 | 0.70 | 0.70 | 0.50 | 0.50 | 0.50 | 0.50 |
| SToRe3D-A1/2 | 0.5 | 0.72 | 1.00 | 0.85 | 0.72 | 0.63 | 0.56 | 0.51 | 0.50 | 1.00 |
| SToRe3D-R1/2 | 0.5 | 0.62 | 1.00 | 0.78 | 0.63 | 0.53 | 0.50 | 0.50 | 0.50 | 0.50 |
| ToC3D-faster | 0.3 | 0.55 | 1.00 | 1.00 | 0.50 | 0.50 | 0.40 | 0.40 | 0.30 | 0.30 |
| SToRe3D-A1/3 | 0.3 | 0.61 | 1.00 | 0.79 | 0.61 | 0.48 | 0.38 | 0.32 | 0.30 | 1.00 |
| SToRe3D-R1/3 | 0.3 | 0.46 | 1.00 | 0.69 | 0.48 | 0.34 | 0.30 | 0.30 | 0.30 | 0.30 |
| SToRe3D-A1/10 | 0.1 | 0.50 | 1.00 | 0.73 | 0.50 | 0.33 | 0.20 | 0.13 | 0.10 | 1.00 |
| SToRe3D-R1/10 | 0.1 | 0.31 | 1.00 | 0.61 | 0.33 | 0.16 | 0.10 | 0.10 | 0.10 | 0.10 |
| **ViT-B Models** | | | | | | | | | | |
| ToC3D-fast | 0.5 | 0.70 | 1.00 | 1.00 | 0.70 | 0.50 | 0.50 | 0.50 | | |
| SToRe3D-A1/2 | 0.5 | 0.74 | 1.00 | 0.78 | 0.63 | 0.53 | 0.50 | 1.00 | | |
| SToRe3D-R1/2 | 0.5 | 0.63 | 1.00 | 0.72 | 0.56 | 0.50 | 0.50 | 0.50 | | |
| ToC3D-faster | 0.3 | 0.58 | 1.00 | 1.00 | 0.50 | 0.40 | 0.30 | 0.30 | | |
| SToRe3D-A1/3 | 0.3 | 0.64 | 1.00 | 0.69 | 0.48 | 0.34 | 0.30 | 1.00 | | |
| SToRe3D-R1/3 | 0.3 | 0.48 | 1.00 | 0.61 | 0.38 | 0.30 | 0.30 | 0.30 | | |
| SToRe3D-A1/10 | 0.1 | 0.53 | 1.00 | 0.61 | 0.33 | 0.16 | 0.10 | 1.00 | | |
| SToRe3D-R1/10 | 0.1 | 0.33 | 1.00 | 0.50 | 0.20 | 0.10 | 0.10 | 0.10 | | |

Table 4: Token pruning schedule for model variants and related approach ToC3D (Zhang et al., 2024) including total keep ratio (TKR), mean keep ratio (MKR), and layer keep ratios (LKR) for each global attention layer in the ViT-L backbone.

