# OpenReview forum: "SToRe3D: Sparse Token Relevance in ViTs for Efficient Multi-View 3D Object Detection"
_ICLR.cc/2026/Conference — ICLR 2026 Conference Withdrawn Submission_

### Official Review · Reviewer_QpEr · 2025-10-28

**Soundness:** 2
**Presentation:** 2
**Contribution:** 3
**Rating:** 4
**Confidence:** 4

**Summary:**

The paper targets the latency bottleneck of multi-view 3D object detection with ViT backbones/DETR-style decoders, where dense processing of (i) many image tokens and (ii) many 3D queries causes quadratic attention cost. It proposes SToRe3D, a planner-aligned sparsity framework that (1) jointly scores and prunes both 2D tokens and 3D queries using lightweight mutual 2D–3D relevance heads, and (2) stores filtered embeddings into buffers for possible reactivation at deeper stages, rather than permanently discarding them.

**Strengths:**

1. Clear problem framing (latency under quadratic attention) and a pragmatic solution that impacts both backbone and decoder.
2. Recoverability through store–reactivate is a practical design that mitigates over-pruning failure modes.
3. Planner alignment is operationalized, not just argued—definitions, labels, and metrics (RM/RA) are all consistent.
4. Solid speed–accuracy tradeoffs with real-time throughput at high sparsity (up to ~18 FPS for ViT-L).

**Weaknesses:**

1. Framework figure clarity. The core architecture diagram (Fig. 1 & Fig. 2) appears low-resolution in the PDF, a larger, higher-resolution schematic (with per-stage keep ratios and reactivation paths labeled) would improve readability.
2. Limited qualitative comparisons. Qualitative evidence is sparse (e.g., Fig. 5 shows a single false-negative comparison versus StreamPETR at similar latency). More comparsion should be provided.
3. Table 1 is hard to interpret. It mixes different backbones (R101, ViT-B, ViT-L), image widths (×800, ×1408, ×1600), and sparsity levels in one block, so SToRe3D’s gains are confounded by backbone/resolution changes.

**Questions:**

Why do mAP and NDS move in different directions in Table 2?
How sensitive are results to $H$ and $d_{min}$?

---

### Official Review · Reviewer_Wqed · 2025-10-30

**Soundness:** 2
**Presentation:** 2
**Contribution:** 1
**Rating:** 2
**Confidence:** 4

**Summary:**

This paper introduces SToRe3D, a relevance-aligned sparsity framework that jointly prunes 2D tokens and 3D queries while storing filtered features for selective reuse. To better evaluate relevance-driven efficiency, the authors also propose nuScenes-Relevance (nuScenes-R), a new benchmark focusing on planning-critical agents for real-time multi-view 3D detection.

**Strengths:**

- Figures 1 and 2 are clear and effectively illustrate the motivation and framework design.
- This paper introduces the concept of object relevance, which aligns sparsity with planning through a future interaction corridor in BEV space. Based on this idea, the authors further propose the nuScenes-R benchmark with relevance-based evaluation metrics.
- The SToRe3D achieves solid results on nuScenes mAP and nuScenes-R, validating its effectiveness.

**Weaknesses:**

- Limited FPS Improvement:
Although SToRe3D aims to enhance efficiency via token and query pruning, the actual FPS gain over the ToC3D baseline (Table 1) is minimal.
- Inconsistent Table Formatting:
Table 1 and Table 2 use inconsistent decimal precision, which reduces clarity and professionalism.

- Figure 4: Missing key baselines (e.g., StreamPETR, ToC3D, PETRv2). The x-axis should not use a log scale, and the latency intervals need to be labeled more clearly to show the actual differences between methods.
- Figure 5: The visualization is confusing. The authors should explicitly annotate false negatives (left image) and correct detections within the interaction corridor (right image). More diverse examples would help clarify the qualitative improvements.
- In the Pruning Design section, the authors claim latency reduction, but Table 3 does not include any latency or speed metrics. Including these would make the ablation analysis more convincing.

- The paper contains numerous typos and grammatical errors that hinder readability.
Line 373: “provides am ablation study” → “provides an ablation study”. Line 323: “To the best of our knowledge the first time transformer models of this scale reach real-time on nuScenes” → Missing comma and awkward phrasing. Line 466: “Since nuScenes-R relies on corridor hyperparameters (H, dmin) future work” → Missing comma after “(H, dmin)”.

**Questions:**

N/A

---

### Official Review · Reviewer_8D9F · 2025-10-30

**Soundness:** 2
**Presentation:** 1
**Contribution:** 2
**Rating:** 2
**Confidence:** 2

**Summary:**

This paper introduces SToRe3D, a framework for efficient multi-view 3D object detection using Vision Transformers (ViTs).

The method jointly prunes 2D image tokens and 3D object queries via mutual 2D–3D relevance heads that score elements according to their planning importance.

The authors further propose nuScenes-Relevance (nuScenes-R), a benchmark emphasizing accuracy on planning-critical agents. Experiments on nuScenes demonstrate that SToRe3D achieves up to 3× faster inference than dense ViT baselines.

**Strengths:**

1. Unified sparsity across tokens and queries with mutual relevance heads is a creative and technically sound contribution.

2. Comprehensive experiments across multiple ViT scales and sparsity regimes; consistent improvements over competitive baselines.

3. nuScenes-R offers a principled way to evaluate planner-aligned perception; its release would benefit the field.

**Weaknesses:**

1. Writing and figure quality need substantial improvement.
a. Figure 1 is visually cluttered and lacks clear legends, making it difficult to interpret the roles of tokens, queries, and buffers.
b. Key terms such as planning-critical agents and store–reactivate are introduced abruptly; they should be explained concisely and intuitively before the formal equations.

2. The paper lacks a per-module latency breakdown, making it unclear where the reported speedup originates. Providing detailed runtime analysis would help substantiate the efficiency claims.

3. The visual examples (Figure 5) are limited and lack clarity. It is not evident where the detection boxes are in the right image. More qualitative results and better annotations would greatly improve interpretability.

Overall, the paper feels rushed and not yet ready for publication

**Questions:**

See weaknesses

---

### Official Review · Reviewer_KjxL · 2025-10-31

**Soundness:** 4
**Presentation:** 4
**Contribution:** 3
**Rating:** 6
**Confidence:** 3

**Summary:**

This work presents SToRe3D, a method to accelerate ViT-based multi-view 3D detection by jointly sparsifying 2D image tokens and 3D object queries.

The proposed method learns to identify and prioritize "planning-relevant" content, using a store-and-reactivate mechanism to avoid irreversible information loss.  Besides, a new benchmark, nuScenes-R, is proposed to measure performance on these critical objects.

**Strengths:**

1. The work addresses the highly relevant problem of inference latency for large-scale ViT-based 3D detectors. The proposed method is intuitive and reasonable.

2. Experiments show that the empirical gains are promising

3. NUSCENES–R is a valuable asset with clear motivations.

**Weaknesses:**

1. The proposed method is very similar to "Towards Efficient Use of Multi-Scale Features in Transformer-Based Object Detectors, CVPR 23". Please discuss your difference with this method. Proper citation might be required.

2. It is difficult to disentangle the performance gains of the novel 'store-reactivate' idea from the general, already-explored strategy of hierarchical token reduction.

**Questions:**

Please see the weaknesses.

---

### Note · Authors · 2025-11-14

I have read and agree with the venue's withdrawal policy on behalf of myself and my co-authors.